# Dietary Pattern Indicators among Healthy and Unhealthy Weight Adolescents Residing in Different Contexts across the Otago Region, New Zealand

**DOI:** 10.3390/children10091445

**Published:** 2023-08-24

**Authors:** Kirsten J. Coppell, Michael Keall, Sandra Mandic

**Affiliations:** 1Department of Medicine, University of Otago Wellington, Wellington South 6242, New Zealand; 2Nelson Marlborough Institute of Technology, Nelson 7010, New Zealand; 3Department of Public Health, University of Otago Wellington, Wellington South 6242, New Zealand; michael.keall@otago.ac.nz; 4School of Sport and Recreation, Faculty of Health and Environmental Sciences, Auckland University of Technology, Auckland 0627, New Zealand; 5Centre for Sustainability, University of Otago, Dunedin 9054, New Zealand; 6AGILE Research Ltd., Wellington 6012, New Zealand

**Keywords:** adolescent, body weight, dietary pattern, neighbourhood characteristics, socioeconomic factors

## Abstract

Reported obesity rates for adolescents in different urban and rural areas are inconsistent. We examined indicators of healthy and unhealthy dietary patterns among 1863 adolescents aged 13–18 years with a healthy or excess body weight attending 23 secondary schools in four different settlement types across the Otago region, New Zealand. An online survey included demographics and dietary behaviours. Height and weight were measured, and body mass index was calculated. New Zealand defined urban and rural settlement types were used. Home addresses determined a small area-level index of deprivation. Data were analysed using Chi-square tests and ANOVA. A logistic model was fitted to estimate adjusted odds ratios of excess weight. The proportion of adolescents with a healthy weight differed (*p* < 0.001) between the most (64.9%) and least (76.4%) deprived neighbourhood areas. There was only indicative evidence of differences between settlement types (*p* = 0.087). Sugar-sweetened beverage and fast-food consumption was more frequent in the most deprived areas (*p* < 0.001), and in urban versus rural settlements (*p* < 0.001). The most important associations with excess weight were area-level deprivation and ethnicity, but not settlement type. Prioritising socioeconomic factors irrespective of settlement type is necessary when developing interventions to improve dietary patterns and body weight status among adolescents.

## 1. Introduction

Excess body weight is common worldwide, including in New Zealand where more than two-thirds of the adult population and one-third of children are overweight or obese as defined by body mass index (BMI) categories [1]. Overweight and obesity can impair health by increasing the risk of non-communicable diseases such as type 2 diabetes, hypertension, non-alcoholic fatty liver disease, osteoarthritis, some cancers and obstructive sleep apnoea [2], some of which are increasingly diagnosed in adolescents [3]. Adults with overweight or obesity also have a higher all-cause mortality compared with healthy weight individuals [2,4]. The World Health Organization reported in 2021 that the global obesity epidemic now causes almost 3 million deaths annually [5].

The cause of overweight and obesity is complex and often stems from childhood and adolescence when health habits or behaviours are often established and track into adulthood [6,7]. While individual factors are important, sociocultural and environmental factors have a significant influence, and are considered to be important in the development of obesity and obesity-related disorders [8,9,10]. The influence of these more distal determinants of health on the development of obesity and non-communicable diseases such as type 2 diabetes [11] are illustrated in models such as the socioecological model [12] and the model of community nutrition [13]. The broader socioecological model recognises five spheres of influence that draws attention to proximal and distal factors (personal, interpersonal, organisational, community and policy), whereas the more specific model of community nutrition environments is a more linear model with three ‘variable’ categories (individual, environmental and policy) that influence eating pattern behaviours. In this latter model, the environmental variable encompasses community, organisational, consumer and information (media and advertising). While it does not specifically incorporate the built environment, this model does refer to some aspects of the built environment that impact obesity risk such as advertising and the location of food outlets and whether food outlets are drive-through.

The built environment refers to the physical and built infrastructure where people live, study, work, play, socialise and travel [14] while the social environment includes the groups to which we belong such as family and the neighbours where we live, the organisation of our workplaces and policies developed to influence people’s lives [15]. Although the built and social environments encompass different aspects of the wider environment, they are not separate entities and components often interact and influence health behaviours [16,17]. For example, engagement in physical activity such as adolescents cycling to school may depend on distance and the availability of safe cycle lanes as well as parental influence [18,19]. While specific community settings such as residential neighbourhoods or school neighbourhoods can have similar built and social environment features, they are typically heterogeneous [17,20]. In addition to geographic variation, community settings can also differ in for example level of rurality, socioeconomic status of residents, housing density, walkability, availability of green spaces for recreation and physical activity, cycling infrastructure, drivability, and food outlet type, location and density. Furthermore, different settings and neighbourhoods can change over time. For example, the number of fast-food outlets that cluster around New Zealand schools in urban areas increased between 1966 and 2006 [21].

Neighbourhood built and social environment characteristics influence obesity risk across the lifespan and are particularly influential during adolescent years [22,23]. This risk is likely to be context specific [24] with some neighbourhoods considered to be less obesogenic, that is, less likely to promote an excess body weight, and others more obesogenic [25]. The exposure and impact of different components of the neighbourhood environment and sociodemographic factors within different geographic areas on eating patterns, obesity risk and obesity rates are not fully understood. This is illustrated by the fact that reported obesity rates for different urban and rural areas are inconsistent. For example, obesity rates have been reported to be higher among adults and adolescents in rural areas compared with urban areas in the US [26,27] and Canada [28], whereas the reverse has been observed in New Zealand and Indonesia where higher obesity rates have been reported among children and adolescents in urban areas compared with rural areas [29,30].

In New Zealand, there has been little focus on obesity and its prevention in adolescents. The prevalence of obesity and overweight and unhealthy dietary habits among adolescents in different urban and rural settings have not been previously examined. The aim of this study was to examine indicators of healthy and unhealthy dietary patterns among adolescents with a healthy body weight or excess body weight attending secondary schools in urban and rural areas across the Otago region, New Zealand.

## 2. Materials and Methods

This cross-sectional study involved secondary analysis of data collected as part of the Built Environment and Active Transport to School (BEATS) Research Programme undertaken in the Otago region, New Zealand. This programme was established in 2013 to examine individual, social, environmental and policy influences on adolescents’ active travel to school in urban and rural areas across the Otago region [31,32]. It also examined other lifestyle factors including body weight and dietary behaviours. The BEATS study was conducted in Dunedin city in 2014–2017 [31] and the BEATS Rural (BEATS-R) study was conducted across the four Otago region districts in 2018 [33]. 

### 2.1. Setting

Otago is the second largest region by land area in New Zealand, measuring approximately 32,000 km^2^, which is about 12% of New Zealand’s total land area. It is a predominantly rural region with one city, Dunedin, and four districts. For this analysis Dunedin city and the neighbouring Mosgiel urban area were considered separately. There are several medium and small urban areas and rural settlements located across the four districts. The usually resident 2018 Census population of the region was 225,186 with ~13% aged 10–19 years [34].

### 2.2. Built Environment and Active Transport to School (BEATS) Research Programme

The methods have been previously described [31,32]. Briefly, adolescents were recruited through all 12 secondary schools in Dunedin city in the BEATS study (n = 1780) and 11 of 15 secondary schools across the Otago region in the BEATS-R study (n = 993). They were given written information about the study and written consent was required prior to participation. In the 2014–2015 BEATS study, depending on participating school’s preference, parental opt-in or opt-out consent was used for adolescents aged <16 years. Parental consent was not required for this age group in the 2018 BEATS-R study. 

An online survey including demographic information and dietary behaviours was completed during school time and supervised by research assistants [31]. Completion of the survey multiple times by a single individual was not possible because schools organised participants to complete their survey during a specific time period in an allocated classroom or a computer laboratory and only present participants with a signed consent form were allocated a unique study ID and given access to the online survey. Demographic characteristics included date of birth, gender, ethnicity and home address. Age was calculated in years at the time of survey. Ethnicity data were coded into five categories using prioritised ethnicity for New Zealand [35]. Home address was used to determine an index of deprivation [36] and whether they lived in an urban or rural setting, as defined below [37].

Dietary behaviours were assessed using questions from the Health Behaviour in School Children survey (permission obtained) (test–retest reliability for Food Frequency Questionnaire for adolescents: Spearman correlations with the 7-day food diary −0.13 to 0.67) [38]. Participants were asked to report their weekly frequency of consuming breakfast, fruits, vegetables, sweets, sugary and soft drinks and fast food using a stem “How many times a week do you usually eat or drink…?”. Response categories were ‘never’, ‘less than once a week’, ‘once a week’, ‘2–4 days a week’, ‘5–6 days a week’, ‘once a day, every day’ and ‘every day, more than once’. These seven response categories were subsequently grouped into three categories (‘once a week or less’, ‘2–4 days a week’ and ‘5 or more days a week’) for analyses as categorical variables. Indicators of a healthy dietary pattern were defined as consuming breakfast, fruits, and vegetables daily and indicators of an unhealthy dietary pattern were consuming sweets, sugary and soft drinks and fast-food regularly.

Height (custom-built portable stadiometer in the BEATS Study; SECA stadiometer in the BEATS Rural Study) and weight (A&D scale UC321, A&D Medical) were measured by trained research assistants using standard procedures described elsewhere [39]. Participants wore their school uniform but removed their shoes and school blazer prior to their height and weight measurements. Body mass index (BMI) was calculated as weight divided by height squared (kg/m^2^). Participants were categorised as ‘underweight’, ‘healthy weight’, ‘overweight’ or ‘obese’ using international age- and sex-specific cut-points for BMI [40].

The New Zealand Index of Deprivation (NZDep) measures the level of socioeconomic deprivation for people resident in each small census area unit (called a meshblock) in New Zealand [36]. It is based on nine Census variables and is updated after each Census. There are 10 deciles with decile 1 representing the least deprived areas and decile 10 representing the most deprived areas. Meshblock codes for adolescents’ home addresses (derived using Geographical Information Systems [31]) were used to determine NZDep deciles. The NZDep was recoded into quintiles: low (1–2), middle-low (3–4), middle (5–6), middle-high (7–8) and high (9–10) deprivation.

The setting of home and school locations were determined by categorising participants’ home and school locations, respectively, into one of the six urban and rural categories defined by Statistics New Zealand (StatsNZ) [37]. This categorisation was applied to an urban-rural zoning used to report 2013 census data that was published on the StatsNZ data portal, as described elsewhere [33]. For this study two of the six urban and rural categories were not used. This was because no area had a population sufficiently large to meet the ‘major urban’ criteria of 100,000 or more. ‘Rural settlements (200–999)’ and ‘other rural areas’ were combined into a single category as the number of participants in each of these two categories was too small for analysis and no secondary schools were located in ‘other rural areas’. The four StatsNZ defined geographical setting categories used were large urban area (30,000–99,999 residents), medium urban area (10,000–29,999 residents, small urban area (1000–9999 residents) and rural setting (<1000 residents) [33]. To limit the confounding of exposure to multiple settlement types, data were analysed only for adolescents who lived and attended a secondary school in the same settlement type (i.e., matched settlement type for home and school location), and not boarding at school or privately.

### 2.3. Data Analysis

Participant data from the BEATS Study and the BEATS-R Study were combined (n = 2773). The analytical sample for this study was 1863 after excluding adolescents for the following reasons: missing written consent (n = 65), invalid (n = 49) or blank (n = 20) survey, invalid home address (n = 5), boarding at school or privately (n = 196), incomplete dietary data (n = 327), BMI was unable to be calculated due to missing height or weight measurements (n = 178) or categorised as underweight (n = 72). 

Data were analysed using Chi-square test for categorical variables and ANOVA with Scheffe post hoc comparisons for continuous variables when homogeneity of variance assumption was met or Tamhane T2 when homogeneity of variance assumption was not met. As various factors may be associated with an unhealthy weight, and these associations are likely to change when the levels of other related factors are controlled for, an attempt was made to disentangle some of these factors by fitting a logistic regression model to estimate the adjusted odds of having an unhealthy weight associated with age group, sex, ethnicity, settlement type and area-level deprivation. Adolescents’ age was included in the model by indicator variables for each year of age from 13 years to 18 years. As students were clustered within schools, an initial generalized linear mixed model was fitted with schools as a random effect. The model could not provide estimates for ethnicity when the random effects were specified, for algorithmic reasons. As the standard errors for other coefficients changed very little when schools were not included as random effects, a simpler model that did not account for clustering (but could estimate coefficients for ethnicity) was preferred. Continuous variables are reported as means and standard deviations (SDs). Categorical variables are reported as frequency and percentage. Odds ratios and 95% confidence intervals are reported. Data were analysed using SPSS Statistical Package version 27.0 and SAS 9.4 (SAS Institute, Cary, NC, USA).

## 3. Results

The demographic characteristics of the study population are shown in Table 1 and Table 2. Overall, more than one-quarter (27%) were either overweight or obese, while more than one-third (35%) of those who resided in a high deprivation area were overweight or obese. Table 1 shows that the proportion of participants in each ethnic group and in each neighbourhood deprivation quintile varied significantly by settlement type (*p* < 0.001). The large urban area had the lowest proportion of adolescents who self-identified as NZ Europeans (72%), the highest proportion who self-identified as ‘Asian’ (7%) and the highest proportion who resided in the most deprived areas (10%) compared with the smaller settlement types. The rural settlements had the highest proportion of adolescents who lived in the least deprived areas, quintiles 1 and 2, (73%) compared with 51–57% in the other three settlement types. The proportion of NZ Europeans in each deprivation quintile decreased with increasing levels of deprivation and the reverse was observed for the other ethnic groups (Table 2). There was only indicative evidence that weight status varied across the settlement types (*p* = 0.087): the large urban area had the lowest proportion of healthy weight adolescents (72%) compared with 76% in the medium urban areas, 77% in the small urban areas and 74% in the rural areas (Table 1). The differences in weight status by area-level deprivation shown in Table 2 were however, statistically significant (*p* < 0.001) with the most deprived area having the lowest proportion of healthy weight adolescents (65%).

As for weight, dietary pattern indicators also varied across deprivation quintiles (Table 3). The differences in sugar sweetened beverage and fast-food consumption between deprivation areas were highly statistically significant. The group who resided in the most deprived areas had the highest proportion of adolescents who consumed sugar sweetened beverages (23%) and fast foods (13%) on most or every day of the week. In contrast to settlement types, statistically significant differences in weekly fruit and vegetable consumption were observed across the different deprivation areas with adolescents residing in the lowest deprivation areas having the lowest reported weekly frequency of fruit and vegetable consumption. Two-thirds (66%) consumed fruit and almost three-quarters (74%) consumed vegetables 5–7 days per week, compared with 79% and 87%, respectively, for adolescents residing in the least deprived areas. A similar pattern was observed for weekday breakfast consumption with statistically significant differences observed among neighbourhood deprivation quintiles.

Only indicators of unhealthy dietary patterns varied significantly across the four study settlement types (Table 4). A higher proportion of adolescents residing in the large urban area reported consuming sweets (23%), sugar sweetened beverages (18%) and fast foods (5%) on most or every day of the week compared with those living in other settlement types. The lowest proportion of adolescents who reported consuming sweets (16%), sugar sweetened beverages (10%) and fast foods (2%) 5–7 days per week were those residing in rural settlements. Overall, less than one-third reported consuming at least one serving of fruit and at least one serving of vegetables each day. For all adolescents across all settlement types combined, 74% reported consuming fruit and 84% reported consuming vegetables on all or most (5–7) days of the week. Although a higher proportion of adolescents in rural settlements consumed fruit (78%) and vegetables (87%) on most (5–7) days of the week differences in fruit and vegetable consumption between settlement types were not significant. Similarly, the consumption of breakfast during the school week did not vary significantly between settlement types.

Table 5 shows the results of the adjusted logistic regression model and the associations of having an unhealthy weight. In this model, living in a high deprivation area, but not settlement type, was associated with an unhealthy weight. 

## 4. Discussion

Addressing risk factors for non-communicable diseases such as type 2 diabetes and non-alcoholic fatty liver disease is an urgent health priority [2,5] especially in childhood and adolescence, a time when lifelong habits are typically established [41]. A poor diet and obesity are well-recognised key risk factors for many common non-communicable diseases, but effective sustainable interventions to address these risk factors are limited because the causes and influences are multifactorial and complex [11,13,17]. This study examined indicators of healthy and detrimental dietary habits among a sample of adolescents with either a healthy or excess weight attending secondary schools in urban and rural areas across the Otago region, New Zealand. Overall, less than three-quarters had a healthy weight. There was little variation in the proportion of adolescents with a healthy weight among settlement types (71.5–77.1%). In contrast, the difference between area levels of deprivation was significant with adolescents residing in the two least deprived neighbourhood quintiles, 1 and 2, having a higher proportion with a healthy weight (76.4% and 78.1%, respectively) than those residing in the two most deprived neighbourhood quintiles, 4 and 5, (67.3% and 64.9%, respectively). There were also marked differences in obesity with 4.1% of adolescents residing in the least deprived areas classified as obese compared with 15.3% in the most deprived areas. A similar pattern was observed for detrimental eating habits with reported weekly sugar sweetened beverage and fast-food consumption being more frequent among adolescents living in the most deprived areas compared with the least deprived areas. These detrimental eating habits were also significantly more common among adolescents in the large urban area, compared with rural settlements which had the highest proportions of adolescents residing in the two least deprived neighbourhood quintiles, 1 and 2. Our findings are consistent with the suggestion that sociodemographic factors within geographic areas are important for obesity risk and that features of area level neighbourhood deprivation have a greater influence on eating behaviours and excess body weight risk than settlement type [24]. This may in part explain inconsistent reporting of differences in obesity rates between urban and rural areas with some studies reporting higher obesity rates in urban areas and others reporting higher rates in rural areas [26,27,28,29,30].

In New Zealand very few studies have examined body weight and dietary habits specifically in adolescents, despite global concerns and recognition of the importance of adolescence health [41], particularly as habits established during adolescence and obesity typically track into adulthood and influence future health [42]. The annual health surveys in New Zealand do not report body size data for New Zealand adolescents, but for children aged 10–14 years and young adults aged 15–24 years [1]. The Youth Survey series provides the only comprehensive nationally representative health and weight and weight-related data for New Zealand adolescents who attend secondary school [43]. They reported the prevalence of excess body weight among adolescents was nearly 40% in 2012 and that, as observed in our study, the prevalence of excess weight was highest among adolescents living in high deprivation areas (49.9%) compared with low deprivation areas (26.9%), and there were few or no differences by sex or settlement type (major city, small city or rural) [43]. The Youth Survey does not however, report eating patterns.

Frequent consumption of sugar sweetened beverages and fast foods are well known to be associated with obesity [44]. In this Otago region study, consuming sweets, sugar sweetened beverages and fast foods most days of the week was significantly more common among adolescents in the large urban area compared with the other three settlement types. Fast food consumption is also more common among New Zealand adults living in urban areas compared with rural areas [45]. Urbanisation is associated with greater access and availability of sugar sweetened beverages and unhealthy fast foods [44], especially in more deprived areas [46] and around schools [21,46]. In New Zealand, schools in the most deprived areas have the greatest density of food outlets compared with those in less deprived areas in the major urban areas [46], and the number of fast-food outlets clustering around schools in urban areas has increased over time [21], despite concerns expressed by parents and schools (www.stuff.co.nz/the-press/news/82991038/christchurch-city-council-will-investigate-limiting-fast-food-outlets-near-schools, accessed on 17 August 2023; www.newshub.co.nz/home/lifestyle/2021/05/health-experts-concerned-by-arrival-of-christchurch-s-first-taco-bell-the-5th-fast-food-chain-built-across-from-primary-school.html, accessed on 17 August 2023). An increase in the number of fast-food outlets around schools over time has been observed elsewhere, for example, in the US [47]. Also of relevance, is that the type of food outlets has changed over time, for example, in Flanders, Belgium where more healthy traditional food stores such as greengrocers, bakeries and butchers have been replaced by less healthy options such as convenience stores, fast food outlets and confectionary stores [48].

Overall, less than one-third of adolescents reported consuming at least one serving of fruit and at least one serving of vegetables daily and the proportion of adolescents who reported eating fruit and vegetables most days of the week differed significantly by area level deprivation. A substantially higher proportion of those living in the least deprived areas compared with those living in the most deprived areas reported eating fruit and vegetables most days. The lower intake of fruit and vegetables among adolescents residing in the more deprived neighbourhood areas irrespective of settlement type is consistent with prior studies among adolescents and adults [43]. While there were no statistically significant differences in fruit and vegetable consumption across settlement types, a slightly higher proportion of adolescents in rural areas reported eating fruit (78%) and vegetables (87%) most days of the week compared with the large urban centre dwellers (75% and 82%, respectively). This pattern has been observed elsewhere and may reflect the fact that fresh fruit and vegetables are more accessible in the central rural part of the Otago region, which is renowned for growing fruit (especially) and vegetables (centralotagonz.com/discover/our-stories/central-otago-fruit-story/, accessed on 17 August 2023). While not typical, a higher vegetable consumption among 15-year-olds has also been observed in remote rural areas of Scotland compared with main urban areas [49]. 

Taken together our findings support the view that contextual neighbourhood factors appear to be of central importance when considering the impact of the built and social environment features on obesity risk, obesity rates and obesity prevention [17,20]. The implication is that it cannot be assumed that features of one urban setting are the same as another, or that features of, for example, areas categorised as the most deprived in an urban setting are the same as areas categorised as the most deprived in a less urban or rural settings [50]. In another words, obesogenic factors identified in one type of setting cannot necessarily be generalised to another type of setting, and that features of one rural or urban setting may not be the same as another rural or urban setting. In this regional study, while obesity rates were similar for the large urban area and rural settlements, availability and consumption of sugar sweetened beverages and fast foods were significantly different, being higher among adolescents in the large urban area compared with those in the rural settlements, suggesting other factors like physical activity levels, household income or neighbourhood deprivation are more influential in rural settlements in Otago [33,34]. 

This study did not specifically examine physical activity, but how adolescents travelled to school whether by active transport or motorised transport or a combination, and their physical activity levels through sport and other activities [33] will also influence obesity risk, and this will also in part be mediated by neighbourhood built environment features such as access to green spaces, safe cycle ways, and presence of footpaths and pedestrian crossings [8,16]. Further, while neighbourhoods contain health promoting features, such as safe walkways, or health constraining features, such a high density of fast-food outlets, or both [51], individual and family factors are also likely to contribute to dietary habits and obesity risk [17]. Although we did not assess these personal and interpersonal factors in the present study, proximal factors are likely to interact with the wider environment and influence obesity risk [52].

### Strengths and Limitations

This study was conducted in different sized urban and rural settlements across an entire region in New Zealand, and a high number of adolescents from 21 of 25 secondary schools across the region participated. However, there are study limitations. The cross-sectional study design precludes inferences about causality. No information was collected on those adolescents who chose not to participate, and it is possible that as a group these adolescents differed in some way. Similarly, those who had incomplete dietary data or did not have both a height and weight measurement recorded and were excluded from this analysis may also have differed in some aspect. While the focus of this paper was dietary patterns the absence of measured physical activity data in this analysis is another limitation. Finally, data were collected during 2014–2018 and therefore do not reflect more recent global changes that may influence dietary patterns, such as the emergence of the COVID-19 pandemic in 2020. 

## 5. Conclusions

Excess body weight is common among adolescents in both urban and rural geographical settings in New Zealand, and more so among adolescents residing in the more deprived areas compared with the less deprived areas. The lower than recommended consumption of fruit and vegetables and the frequent consumption of sugar sweetened beverages and fast foods was also common, and like excess body weight, these indicators of an unhealthy dietary pattern were more common in the more deprived areas compared with the less deprived areas. Except for the more frequent consumption of sugar sweetened beverages and fast foods in the large urban area compared with the smaller urban and rural areas, there were no significant differences in body weight and fruit and vegetable consumption between geographic settlements. These observations support the notion that neighbourhood deprivation is a key driver of obesity risk and obesity, and different geographical settings are less influential. Effective interventions to improve eating patterns and reduce rates of excess body weight among adolescents are urgently needed and these need to consider and prioritise socioeconomic factors irrespective of geographical setting.

## Figures and Tables

**Table 1 children-10-01445-t001:** The sociodemographic characteristics and weight status of adolescents by settlement type.

Sociodemograpic Characteristics	Total n = 1863	Large Urban n = 875	Medium Urban n = 190	Small Urban n = 284	Rural n = 514	*p*-Value
Age (years), mean (SD)	15.3 (1.4)	15.3 (1.5)	15.4 (1.4)	15.3 (1.3)	15.2 (1.4)	0.568
Gender, n (%)						
Boys	878 (47.1%)	397 (45.4%)	95 (50.0%)	134 (47.2%)	252 (49.0%)	0.483
Girls	985 (52.9%)	478 (54.6%)	95 (50.0%)	150 (52.8%)	262 (51.0%)	
Ethnicity ^1^, n (%)						
NZ European	1384 (74.6%)	624 (71.6%)	150 (78.9%)	210 (73.9%)	400 (78.3%)	<0.001
Māori	218 (11.7%)	103 (11.8%)	15 (7.9%)	36 (12.7%)	64 (12.5%)	
Pacific	68 (3.7%)	37 (4.2%)	8 (4.2%)	7 (2.5%)	16 (3.1%)	
‘Asian’	84 (4.5%)	63 (7.2%)	5 (2.6%)	8 (2.8%)	8 (1.6%)	
Other	102 (5.5%)	44 (5.1%)	12 (6.3%)	23 (8.1%)	23 (4.5%)	
NZDep quintile ^2,3^, n (%)						
1 (least deprived)	564 (31.2%)	255 (29.6%)	47 (28.8%)	67 (23.9%)	195 (38.9%)	<0.001
2	485 (26.9%)	194 (22.5%)	46 (28.2%)	75 (26.8%)	170 (33.9%)	
3	385 (21.3%)	186 (21.6%)	32 (19.6%)	89 (31.8%)	78 (15.6%)	
4	260 (14.4%)	142 (16.5%)	27 (16.6%)	36 (12.9%)	55 (11.0%)	
5 (most deprived)	111 (6.1%)	84 (9.8%)	11 (6.7%)	13 (4.6%)	3 (0.6%)	
Weight status ^4^, n (%)						
Healthy weight	1368 (73.4%)	626 (71.5%)	145 (76.3%)	219 (77.1%)	378 (73.5%)	0.087
Overweight	365 (19.6%)	179 (20.5%)	32 (16.8%)	57 (20.1%)	97 (18.9%)	
Obese	130 (7.0%)	70 (8.0%)	13 (6.8%)	8 (2.8%)	7.6%)	

Abbreviations—SD: Standard Deviation; NZ: New Zealand; NZDep: New Zealand Index of Deprivation. ^1^ Ethnicity missing for 3 rural participants. ^2^ NZDep measures the level of socioeconomic deprivation for people resident in each small census area unit in New Zealand [36]. It is based on nine Census variables and is updated after each Census. There are 10 deciles with 1 representing the least deprived areas and 10 representing the most deprived areas. For this study the NZDep was recoded into quintiles: low (1–2), middle-low (3–4), middle (5–6), middle-high (7–8) and high (9–10) deprivation. ^3^ NZDep score missing for 13 rural participants. ^4^ Participants were categorised as ‘healthy weight’, ‘overweight’ or ‘obese’ using international age- and sex-specific cut-points for body mass index (BMI) [40].

**Table 2 children-10-01445-t002:** The sociodemographic characteristics and weight status of adolescents by neighbourhood area-level socioeconomic deprivation quintile.

Characteristics	Total	New Zealand Index of Deprivation (NZDep) Quintile ^1^					
	n = 1805	1 (Least Deprived) n = 564	2 n = 485	3 n = 385	4 n = 260	5 (Most Deprived) n = 111	*p*-Value
Age (years), mean (SD)	15.3 (SD 1.4)	15.3 (SD 1.4)	15.3 (SD 1.5)	15.4 (SD 1.4)	15.3 (SD 1.5)	15.3 (SD 1.5)	0.886
Gender, n (%)							
Boys	862 (47.8%)	279 (49.5%)	255 (52.6%)	172 (44.7%)	118 (45.4%)	38 (34.2%)	0.004
Girls	943 (52.2%)	285 (50.5%)	230 (47.4%)	213 (55.3%)	142 (54.6%)	73 (65.8%)	
Ethnicity ^2^, n (%)							
NZ European	1343 (74.7%)	456 (81.0%)	378 (78.1%)	271 (70.9%)	165 (63.7%)	73 (65.8%)	<0.001
Māori	212 (11.8%)	41 (7.3%)	52 (10.7%)	61 (16.0%)	42 (16.2%)	16 (14.4%)	
Pacific	64 (3.6%)	8 (1.4%)	14 (2.9%)	16 (4.2%)	16 (6.2%)	10 (9.0%)	
‘Asian’	82 (4.6%)	9 (3.4%)	17 (3.5%)	13 (3.4%)	27 (10.4%)	6 (5.4%)	
Other	98 (5.4%)	39 (6.9%)	23 (4.8%)	21 (5.5%)	9 (3.5%)	6 (5.4%)	
Weight status ^3^, n (%)							
Healthy weight	1332 (73.8%)	431 (76.4%)	379 (78.1%)	275 (71.4%)	175 (67.3%)	72 (64.9%)	<0.001
Overweight	345 (19.1%)	110 (19.5%)	80 (16.5%)	77 (20.0%)	56 (21.5%)	22 (19.8%)	
Obese	128 (7.1%)	23 (4.1%)	26 (5.4%)	33 (8.6%)	29 (11.2%)	17 (15.3%)	

Abbreviations—SD: Standard Deviation; NZ: New Zealand. ^1^ NZDep measures the level of socioeconomic deprivation for people resident in each small census area unit in New Zealand [36]. It is based on nine Census variables and is updated after each Census. There are 10 deciles with 1 representing the least deprived areas and 10 representing the most deprived areas. For this study the NZDep was recoded into quintiles: low (1–2), middle-low (3–4), middle (5–6), middle-high (7–8) and high (9–10) deprivation. ^2^ Ethnicity data missing for 6 participants. ^3^ Participants were categorised as ‘healthy weight’, ‘overweight’ or ‘obese’ using international age- and sex-specific cut-points for body mass index (BMI) [40].

**Table 3 children-10-01445-t003:** Adolescents self-reported weekly consumption of different foods by neighbourhood area-level socioeconomic deprivation quintile.

Number of Days per Week That Different Foods or Meals Were Consumed	Neighbourhood Area-Level Socioeconomic Deprivation (NZDep ^1^) Quintile					
	1 (Least) n = 564 n (%)	2 n = 485 n (%)	3 n = 385 n (%)	4 n = 260 n (%)	5 (Most) n = 111 n (%)	*p*-Value
Fruit, n (%)						
0–1	40 (7.1%)	33 (6.8%)	34 (8.8%)	31 (11.9%)	12 (10.8%)	0.032
2–4	78 (13.8%)	85 (17.5%)	65 (16.9%)	46 (17.7%)	26 (23.4%)	
5–7	446 (79.1%)	367 (75.7%)	286 (74.3%)	183 (70.4%)	73 (65.8%)	
Vegetables, n (%)						
0–1	14 (2.5%)	13 (2.7%)	20 (5.2%)	16 (6.2%)	8 (7.2%)	0.001
2–4	61 (10.8%)	49 (10.1%)	43 (11.2%)	40 (15.4%)	21 (18.9%)	
5–7	489 (86.7%)	423 (87.2%)	322 (83.6%)	204 (78.5%)	82 (73.9%)	
Sweets, n (%)						
0–1	221 (39.2%)	205 (42.3%)	162 (42.1%)	104 (40.0%)	37 (33.3%)	0.349
2–4	229 (40.6%)	186 (38.4%)	145 (37.7%)	102 (39.2%)	40 (36.0%)	
5–7	114 (20.2%)	94 (19.4%)	78 (20.3%)	54 (20.8%)	34 (30.6%)	
Sugar sweetened beverages, n (%)						
0–1	367 (65.1%)	324 (66.8%)	245 (63.6%)	142 (54.6%)	53 (47.7%)	0.001
2–4	128 (22.7%)	98 (20.2%)	81 (21.0%)	69 (26.5%)	32 (28.8%)	
5–7	69 (12.2%)	63 (13.0%)	59 (15.3%)	49 (18.8%)	26 (23.4%)	
Fast foods, n (%)						
0–1	506 (83.9%)	416 (81.8%)	315 (81.1%)	203 (78.1%)	78 (70.3%)	<0.001
2–4	43 (12.1%)	55 (12.8%)	52 (15.3%)	49 (18.8%)	19 (17.1%)	
5–7	15 (4.0%)	14 (5.4%)	18 (2.7%)	8 (3.1%)	14 (12.6%)	
Breakfast consumed on weekdays, n (%)						
0	61 (10.8%)	49 (10.1%)	68 (17.7%)	45 (17.3%)	26 (23.4%)	<0.001
1	18 (3.2%)	18 (3.7%)	18 (4.7%)	13 (5.0%)	6 (5.4%)	
2	22 (3.9%)	27 (5.6%)	22 (5.7%)	28 (10.8%)	3 (2.7%)	
3	36 (6.4%)	34 (7.0%)	31 (8.1%)	22 (8.5%)	7 (6.3%)	
4	42 (7.4%)	42 (8.7%)	38 (9.9%)	24 (9.2%)	12 (10.8%)	
5	385 (68.3%)	315 (64.9%)	208 (54.0%)	128 (49.2%)	57 (51.4%)	
Consumed at least one serving of fruit and at least one serving of vegetables daily, n (%)	193 (34.2%)	140 (28.9%)	116 (30.1%)	60 (23.1%)	20 (18.0%)	0.001

^1^ NZDep measures the level of socioeconomic deprivation for people resident in each small census area unit in New Zealand [36]. It is based on nine Census variables and is updated after each Census. There are 10 deciles with 1 representing the least deprived areas and 10 representing the most deprived areas. For this study the NZDep was recoded into quintiles: low (1–2), middle-low (3–4), middle (5–6), middle-high (7–8) and high (9–10) deprivation.

**Table 4 children-10-01445-t004:** Adolescents self-reported weekly consumption of different foods by settlement type.

Number of Days per Week That Different Foods or Meals Were Consumed	Total n = 1863	Large Urban n = 875	Medium Urban n = 190	Small Urban n = 284	Rural n = 514	*p*-Value
Fruit, n (%)						
0–1	160 (8.6%)	74 (8.5%)	21 (11.1%)	29 (10.2%)	36 (7.0%)	0.331
2–4	312 (16.7%)	149 (17.0%)	33 (17.4%)	53 (18.7%)	77 (15.0%)	
5–7	1391 (74.7%)	652 (74.5%)	136 (71.6%)	202 (71.1%)	401 (78.0%)	
Vegetables, n (%)						
0–1	77 (4.1%)	42 (4.8%)	10 (5.3%)	10 (3.5%)	15 (2.9%)	0.283
2–4	221 (11.9%)	115 (13.1%)	23 (12.1%)	31 (10.9%)	52 (10.1%)	
5–7	1565 (84.0%)	718 (82.1%)	157 (82.6%)	243 (85.6%)	447 (87.0%)	
Sweets, n (%)						
0–1	761 (40.8%)	353 (40.3%)	83 (43.7%)	103 (36.3%)	222 (43.2%)	0.025
2–4	718 (38.5%)	320 (36.6%)	72 (37.9%)	115 (40.5%)	211 (41.1%)	
5–7	384 (20.6%)	202 (23.1%)	35 (18.4%)	66 (23.2%)	81 (15.8%)	
Sugar sweetened beverages, n (%)						
0–1	1171 (62.9%)	496 (56.7%)	114 (60.0%)	194 (68.3%)	367 (71.4%)	<0.001
2–4	414 (22.2%)	220 (25.1%)	47 (24.7%)	52 (18.3%)	95 (18.5%)	
5–7	278 (14.9%)	159 (18.2%)	29 (15.3%)	38 (13.4%)	52 (10.1%)	
Fast foods, n (%)						
0–1	1563 (83.9%)	716 (81.8%)	154 (81.1%)	238 (83.8%)	455 (88.5%)	0.011
2–4	226 (12.1%)	112 (12.8%)	29 (15.3%)	36 (12.7%)	49 (9.5%)	
5–7	74 (4.0%)	47 (5.4%)	7 (2.7%)	10 (3.5%)	10 (1.9%)	
Breakfast consumed on weekdays, n (%)						
0	258 (13.8%)	123 (14.1%)	21 (11.1%)	32 (11.3%)	82 (16.0%)	0.075
1	77 (4.1%)	35 (4.0%)	6 (3.2%)	13 (4.6%)	23 (4.5%)	
2	106 (5.7%)	56 (6.4%)	8 (4.2%)	21 (7.4%)	21 (4.1%)	
3	136 (7.3%)	58 (6.6%)	20 (10.5%)	31 (10.9%)	27 (5.3%)	
4	163 (8.7%)	76 (8.7%)	20 (10.5%)	26 (9.2%)	41 (8.0%)	
5	1123 (60.3%)	527 (60.2%)	115 (60.5%)	161 (56.7%)	320 (62.3%)	
Consumed at least one serving of fruit and at least one serving of vegetables daily, n (%)	546 (29.3%)	253 (28.9%)	58 (30.5%)	82 (28.9%)	153 (29.8%)	0.964

**Table 5 children-10-01445-t005:** The associations of excess body weight with sociodemographic variables, settlement type and neighbourhood area-level socioeconomic deprivation.

Characteristic	Adjusted OR ^1^	95% CI
**Gender**		
Boys	ref	
Girls	0.99	(0.79, 1.22)
**Age** (**years**)		
13	0.87	(0.52, 1.47)
14	0.85	(0.55, 1.31)
15	0.90	(0.58, 1.40)
16	0.83	(0.52, 1.31)
17	1.10	(0.69, 1.74)
18	ref	
**Ethnicity**		
NZ European	ref	
Māori	1.48	(1.08, 2.03)
Pacific	2.67	(1.57, 4.51)
Asian	0.69	(0.39, 1.22)
Other	0.73	(0.44, 1.23)
**Settlement type**		
Rural (<1000 residents)	ref	
Small urban (1000–9999 residents)	0.81	(0.57, 1.15)
Medium urban (10,000–29,999 residents)	0.73	(0.47, 1.12)
Large urban (≥30,000 residents)	1.02	(0.78, 1.32)
**NZDep ^2^ quintile**		
1 (least deprived)	ref	
2	0.89	(0.66, 1.19)
3	1.25	(0.90, 1.70)
4	1.48	(1.06, 2.07)
5 (most deprived)	1.58	(1.00, 2.49)

Abbreviations—CI: Confidence interval; OR: Odds ratio; NZ: New Zealand; NZDep: New Zealand Index of Deprivation. ^1^ Each adjusted odds ratio is adjusted for levels of all other variables in this table. ^2^ NZDep measures the level of socioeconomic deprivation for people resident in each small census area unit in New Zealand [36]. It is based on nine Census variables and is updated after each Census. There are 10 deciles with 1 representing the least deprived areas and 10 representing the most deprived areas. For this study the NZDep was recoded into quintiles: low (1–2), middle-low (3–4), middle (5–6), middle-high (7–8) and high (9–10) deprivation.

## Data Availability

The data presented in this study are not publicly available due to data confidentiality and participants having been given assurances that the collected data will not be shared.

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
