# Peer review of "Dietary Pattern Indicators among Healthy and Unhealthy Weight Adolescents Residing in Different Contexts across the Otago Region, New Zealand"

_children, 2023, doi:10.3390/children10091445_

Round 1

Reviewer 1 Report

Dear authors, 

I have reviewed with interest your paper. The topic, exploring indicators for obesity and overweight risk in adolescents living in different contexts, is of high interest, not only from a scientific point of view, but also from a public health perspective. 

The paper is well written. Introduction is complete, aims of the study are clearly expressed. Material and methods section is also complete. 

As for the result section: 

- line 234: Table 3 is indicated but you are reporting results of the association between dietary pattern indicators and deprivation quintiles which is instead what is reported in table 4

- line 257: the opposite, you indicated table 4 but you are reporting results referred to Table 3

- line 262-263: I am not sure I would include this statement considering that you did not make a comparison between the frequencies of consumption you have found in your survey respect to what is reccommended by scientific societies 

Discussion is also well-written and complete:

- line 284-285: correct "among children and adolescents" with " in childhood and adolescence" 

- line 345 the same as for lines 262-263

- lines 376-386: consider to include the absence of data on physical activity among the limitations of the study

Author Response

Response to Reviewer 1 Comments

Point 1: I have reviewed with interest your paper. The topic, exploring indicators for obesity and overweight risk in adolescents living in different contexts, is of high interest, not only from a scientific point of view, but also from a public health perspective. 

The paper is well written. Introduction is complete, aims of the study are clearly expressed. Material and methods section is also complete. 

Response 1: We thank the reviewer for their positive comments, including the fact that our paper is of high interest from a scientific point of view and from a public health perspective.

As for the result section: 

Point 2: - line 234: Table 3 is indicated but you are reporting results of the association between dietary pattern indicators and deprivation quintiles which is instead what is reported in table 4

Response 2: Thank-you for noting this error. Tables 3 and 4 have been swapped, and the table numbers corrected accordingly.

Point 3: - line 257: the opposite, you indicated table 4 but you are reporting results referred to Table 3

Response 3: Thank-you for noting this error. Tables 3 and 4 have been swapped, and the table numbers corrected accordingly.

Point 4: - line 262-263: I am not sure I would include this statement considering that you did not make a comparison between the frequencies of consumption you have found in your survey respect to what is reccommended by scientific societies 

Response 4: We agree that our study did not include a comparison with fruit and vegetable intake recommendations. Therefore, this sentence (The reported weekly frequency of fruit and vegetable consumption was less than recommended.) has been replaced with our study observations, ‘Overall, less than one-third reported consuming at least one serving of fruit and at least one serving of vegetables each day.’ (lines 278-279).

Point 5: Discussion is also well-written and complete:

Response 5: Thank-you.

Point 6: - line 284-285: correct "among children and adolescents" with " in childhood and adolescence" 

Response 6: This has been corrected (lines 312-313).

Point 7: - line 345 the same as for lines 262-263

Response 7: We agree with this comment that our study did not include a comparison with fruit and vegetable intake recommendations, and therefore the sentence has been modified to describe what was observed in our study. The first part of the sentence (‘Overall, the consumption of fruit and vegetables was less frequent than recommended…’) has been replaced by ‘Overall, less than one-third of adolescents reported consuming at least one serving of fruit and at least one serving of vegetables daily…’) (lines 376-377).

Point 8: - lines 376-386: consider to include the absence of data on physical activity among the limitations of the study

Response 8: Thank-you for making this point. We have added the following sentence to the Strengths and Limitations subsection: ‘While the focus of this paper was dietary patterns the absence of measured physical activity data in this analysis is another limitation.’ (lines 431-433).

Reviewer 2 Report

The paper is well written and the information can be used as good updated statistics for related community.

I recommend to accept the paper in current form 

Minor editing

Author Response

Response to Reviewer 2 Comments

Point 1: The paper is well written and the information can be used as good updated statistics for related community.

I recommend to accept the paper in current form 

Response 1: We thank this reviewer for reviewing our manuscript and providing positive feedback.

As the reviewer's recommendation was 'I recommend to accept the paper in current form' we have not made any changes in relation to this review.

Point 2: Comments on the Quality of English Language

Minor editing

Response 2: We have proofread our manuscript and made occasional edits.

Reviewer 3 Report

Manuscript ID: children-2508254

Dietary Pattern Indicators Among Healthy and Unhealthy Weight Adolescents Residing in Different Contexts Across the Otago Region, New Zealand

Comments and suggestions

1.       The abstract does not mention a clear objective.

2.       Regarding the main findings on sugar sweetened beverage and fast-food consumption was more frequent in the most deprived areas and in urban versus rural settlements. Could the authors explain the reasons as well as mentioned the variables that were controlled to prevent bias.

3.       Respect to the associations with excess weight founded with area-level deprivation and ethnicity, but not settlement type, what were the attributable causes and necessary causes?

4.       The authors conclude about that is necessary prioritizing socioeconomic factors irrespective of settlement type is necessary when developing interventions to improve dietary patterns and body weight status among adolescents, but in their experience and based on evidence, could expand this recommendation further. Since the way in which it is concluded does not contribute something new, but if the authors with their findings can share a methodology, indicators, that is, a structured recommendation, the study could be before socially transferable evidence.

5.       Se recomienda integrar los objetivos particulares o específicos, ya que al inicio sólo se menciona un objetivo general o principal, pero se realizaron otros cálculos.

6.       Se recomienda integrar in diagrama que muestre los criterios que se siguieron con la muestra.

7.       Debe haber un apartado de cuáles fueron los criterios de inclusión y no inclusión.

8.       El tamaño de la muestra y su cálculo no es claro, considerando los análisis mencionados en el apartado de estadística.

9.       The definition of main variables as healthy and unhealthy aren´t mentioned and the main objective was to examine the indicators of healthy and unhealthy dietary patterns among adolescents with a healthy body weight or excess body weight.

10.    The results in the tables when comparing by ethnicity as well as other socioeconomic variables, do not mention how the analyzes were standardized, since the sample sizes by type of settlement are different.

11.    The authors observed as limitations of the study, but it is necessary to mention the situation of the environment of these young people, for example, marital status of the parents, number of siblings, level of education of parents, measuring food security in homes, food culture, etc. In such a way that it is possible to approximate the conclusions made by the authors.

12.    Regarding the OR analysis, it is recommended to carry out an analysis for the variables that were significant and with values inside confidence intervals.

13.    According to this objective and the main outcomes: “The aim of this study was to examine indicators of healthy and unhealthy dietary patterns among adolescents with a healthy body weight or excess body weight attending secondary schools in urban and rural areas across the Otago region, New Zealand”, please share could the authors share these findings that solve or contribute to the identified problem. Could the authors share these findings that solve or contribute to the identified problem. That is, can the transfer of any solution derived from these findings occur? Could the authors explain a bit more?

Author Response

Point 1: The abstract does not mention a clear objective.

Response 1: We undertook an observational study and the objective of the study was to examine indicators of healthy and unhealthy dietary patterns among 1863 adolescents aged 13-18 years with a healthy or excess body weight attending 23 secondary schools in four different settlement types across the Otago region, New Zealand. We used the phrase ‘We examined….’ rather than ‘The objective of this study was to examine…..’ so that we could meet the journal’s specified word count limit for the abstract. Further, the current wording is also a common way of reporting research objective(s) in abstracts written in the English language. Therefore, we have not changed the wording in the abstract.

Point 2: Regarding the main findings on sugar sweetened beverage and fast-food consumption was more frequent in the most deprived areas and in urban versus rural settlements. Could the authors explain the reasons as well as mentioned the variables that were controlled to prevent bias.

Response 2: As noted above we undertook an observational cross-sectional study, and thus, we were not able to make statements relating to causality or reasons for our observations. As is desirable when analysing associations in cross-sectional data, we controlled for characteristics that could potentially confound the relationships we were examining in the model that was fitted. More specifically, we used a logistic regression model to estimate the adjusted odds of having an unhealthy weight associated with age group, sex, ethnicity, settlement type and area-level deprivation (see section 2.2 Data analysis). This meant, for example, that the effects of differences in deprivation levels across the four different settlement types could be controlled for.

Point 3: Respect to the associations with excess weight founded with area-level deprivation and ethnicity, but not settlement type, what were the attributable causes and necessary causes?

Response 3: Thank you for this question. As we undertook a cross-sectional observational study, we were unable to determine the attributable and necessary causes of the associations between excess weight and area-level deprivation and ethnicity. This would have required a different study design and the study would have needed to be large enough with a sufficient sample size in each of the ethnic groups of interest to be able to answer this question. This type of analysis was beyond the scope of our study, and we note this as a limitation of our study – ‘The cross-sectional study design precludes inferences about causality.’ (lines 422-423)

Point 4: The authors conclude about that is necessary prioritizing socioeconomic factors irrespective of settlement type is necessary when developing interventions to improve dietary patterns and body weight status among adolescents, but in their experience and based on evidence, could expand this recommendation further. Since the way in which it is concluded does not contribute something new, but if the authors with their findings can share a methodology, indicators, that is, a structured recommendation, the study could be before socially transferable evidence.

Response 4: Our conclusion is based on our study findings. Understanding obesity is complex and in this study, we wanted to gain a better understanding of why reported obesity rates for different urban and rural areas are inconsistent and whether this is related to an urban or rural environment or levels of deprivation irrespective of whether one lives in an urban or rural environment as explained in the Introduction section (lines 88-98). We consider that this topic is important to understand so that effective obesity prevention interventions are implemented that are effectively tailored to a particular place.

Point 5: Se recomienda integrar los objetivos particulares o específicos, ya que al inicio sólo se menciona un objetivo general o principal, pero se realizaron otros cálculos.

Response 5: We apologise that as this comment is not in the English language, we do not understand, and are therefore unable to respond.

Point 6: Se recomienda integrar in diagrama que muestre los criterios que se siguieron con la muestra.

Response 6: We apologise that as this comment is not in the English language, we do not understand, and are therefore unable to respond.

Point 7: Debe haber un apartado de cuáles fueron los criterios de inclusión y no inclusión.

Response 7: We apologise that as this comment is not in the English language, we do not understand, and are therefore unable to respond.

Point 8: El tamaño de la muestra y su cálculo no es claro, considerando los análisis mencionados en el apartado de estadística.

Response 8: We apologise that as this comment is not in the English language, we do not understand, and are therefore unable to respond.

Point 9: The definition of main variables as healthy and unhealthy aren´t mentioned and the main objective was to examine the indicators of healthy and unhealthy dietary patterns among adolescents with a healthy body weight or excess body weight.

Response 9: Thank for this useful point, which we agree with. We have now added the definitions for indicators of healthy and unhealthy dietary patterns, which reads ‘Indicators of a healthy dietary pattern were defined as consuming breakfast, fruits, and vegetables daily and indicators of an unhealthy dietary pattern were consuming sweets, sugary and soft drinks and fast-food regularly.’ (lines 147-150). 

Point 10: The results in the tables when comparing by ethnicity as well as other socioeconomic variables, do not mention how the analyzes were standardized, since the sample sizes by type of settlement are different.

Response 10: We acknowledge that the number of respondents in each of the four settlement types were different. However, we did not standardise our analyses by age or gender as we did not expect there to be any differences, and indeed our analysis of the data confirmed that there were no differences in age and gender between the four settlement types. The other data and measures were variables of interest. In addition, as mentioned above (Response 2), we undertook a logistic regression analysis and the results of this analysis are reported in Table 5. This analysis addressed potential issues meaning the groups of students were comparable, as each adjusted odds ratios is adjusted for levels of all other variables in this table.

Point 11: The authors observed as limitations of the study, but it is necessary to mention the situation of the environment of these young people, for example, marital status of the parents, number of siblings, level of education of parents, measuring food security in homes, food culture, etc. In such a way that it is possible to approximate the conclusions made by the authors.

Response 11: We acknowledge that it would be useful to have this additional information to provide more detailed analyses. The collection of such information was beyond the scope of this study.   In addition, we wanted to minimise respondent burden and compromising responses by having too many survey questions.

Point 12: Regarding the OR analysis, it is recommended to carry out an analysis for the variables that were significant and with values inside confidence intervals.

Response 12: Thank you for this comment. We included variables that have been shown in the literature to have an association with weight status, whatever their level of significance in the model fitted in our study. This follows an approach to building such models commonly used in public health research (see https://academic.oup.com/ejcts/article/55/2/179/5265263 for a more detailed explanation).

Point 13: According to this objective and the main outcomes: “The aim of this study was to examine indicators of healthy and unhealthy dietary patterns among adolescents with a healthy body weight or excess body weight attending secondary schools in urban and rural areas across the Otago region, New Zealand”, please share could the authors share these findings that solve or contribute to the identified problem. Could the authors share these findings that solve or contribute to the identified problem. That is, can the transfer of any solution derived from these findings occur? Could the authors explain a bit more?

Response 13: We would sincerely like to share solutions or interventions to address the global issue of obesity. However, the causes of obesity are complex and similarly the solutions are complex and we do not have a particular solution to share. By undertaking our study, we are contributing knowledge to help design effective solutions for this global health problem. Our key point, which is presented in our manuscript, is that context needs to be considered when designing and implementing obesity prevention interventions for adolescents, and that it seems that overall socioeconomic factors are more important than the geographic factors, but urbanisation needs to be considered.

Reviewer 4 Report

The manuscript entitled “Dietary Pattern Indicators Among Healthy and Unhealthy Weight Adolescents Residing in Different Contexts Across the Otago Region, New Zealand” has aim to examine indicators of dietary patterns (healthy and unhealthy) among adolescents (attending secondary schools) with a normal or excess body weight in urban and rural areas across the Otago region, New Zealand.

General – title is very promising and currently important due to the increasing prevalence of childhood obesity but “Indicators of dietary patterns” authors cannot only refer to poorer and richer areas i.e., urban and rural areas, there are many other factors that influence dietary patterns, which need to be refined so that this title of the manuscript could pass as such.

Also, it should be noted that the data were collected almost 10 years ago, which is a lot in the world of today's global changes related to food and nutrition and the increase in childhood obesity.

Some of the other comments and suggestions for the manuscript:

Title -  what exactly did you mean by the term “contexts”?

Line 33 - Aotearoa New Zealand – do you mean only on the North Island or both because if so, only New Zealand is enough, rest is unnecessary

Lie 75 – “outlets” – some better phrase should be used, e.g., “places”

Line 128/129 – provide reference for the questionnaire used

English proofreading is recommended, because some individual phases and the terminology used are not In accordance with professional language.

Author Response

Response to Reviewer 4 Comments

Point 1: The manuscript entitled “Dietary Pattern Indicators Among Healthy and Unhealthy Weight Adolescents Residing in Different Contexts Across the Otago Region, New Zealand” has aim to examine indicators of dietary patterns (healthy and unhealthy) among adolescents (attending secondary schools) with a normal or excess body weight in urban and rural areas across the Otago region, New Zealand.

General – title is very promising and currently important due to the increasing prevalence of childhood obesity but “Indicators of dietary patterns” authors cannot only refer to poorer and richer areas i.e., urban and rural areas, there are many other factors that influence dietary patterns, which need to be refined so that this title of the manuscript could pass as such.

Response 1: Thank you for agreeing that the increasing prevalence of childhood obesity is a concern. We acknowledge that there are many other factors that influence dietary patterns. However, for this paper we specifically looked at settlement types as some research has concluded that obesity is more common in rural areas and other research has not found this to be the case. This suggests to us that not all rural or urban areas are the same, and that perhaps socioeconomic deprivation is more important than levels of urbanisation.

Point 2: Also, it should be noted that the data were collected almost 10 years ago, which is a lot in the world of today's global changes related to food and nutrition and the increase in childhood obesity.

Response 2: We acknowledge that the data for this research were collected during 2014-2018 as part of a larger research programme (as described in our paper) and that these data may not reflect more recent global changes such as the impact of the COVID-19 pandemic. Nevertheless, we believe that our research provides significant contribution to the discussion of dietary pattern indicators among health and unhealthy weight adolescents living in different settlement types and the importance of the context in which research is conducted. We now acknowledge this as one of the limitations of this research. We have added the following sentence to the Strengths and Limitations subsection: ‘Finally, data were collected during 2014-2018 and therefore do not reflect more recent global changes that may influence dietary patterns, such as the emergence of the COVID-19 pandemic in 2020.’  (lines 433-435).

Point 3: Some of the other comments and suggestions for the manuscript:

Title -  what exactly did you mean by the term “contexts”?

Response 3: We use the term “contexts” as context may include the physical surroundings of topography, movement patterns and infrastructure, built environment, the governance structures, and the cultural, social and economic environment.

Point 4: Line 33 - Aotearoa New Zealand – do you mean only on the North Island or both because if so, only New Zealand is enough, rest is unnecessary

Response 4: We use ‘Aotearoa New Zealand’ to describe our country as this is how our country is now frequently referred to today to reflect the name used by Māori, the indigenous population. Aotearoa New Zealand refers to the whole country, that is, all the islands. Therefore our preference is to use ‘Aotearoa New Zealand’ to describe our country, but if the Editor advises that we should refer to our country as ‘New Zealand’, we will make such a change.

Point 5: Lie 75 – “outlets” – some better phrase should be used, e.g., “places”

Response 5: We have used the term ‘fast-food outlets’ in line 75 (and in other parts of our manuscript), as this is a commonly used phrase in the literature, and we consider this to be an appropriate phrase.

Point 6: Line 128/129 – provide reference for the questionnaire used

Response 6: Reference [38] used in this sentence is the reference for the questionnaire. The details as listed in our references are:

Currie, C.; Nic Gabhainn, S.; Godeau, E. The Health Behaviour in School-aged Children: WHO Collaborative Cross-National (HBSC) study: origins, concept, history and development 1982-2008. Int J Public Health 2009, 54 Suppl 2, 131-139, doi:10.1007/s00038-009-5404-x.

Point 7: Comments on the Quality of English Language

English proofreading is recommended, because some individual phases and the terminology used are not In accordance with professional language.

Response 7: We have proofread our paper and made only occasional minor changes. We agree with Reviewer 2 that our paper is ‘well written’ and minor editing only was suggested. Reviewer 1 considered the ‘English language was fine and no issues were detected’. Of note, the authors are either native English speakers or researchers with over 20 years of experience of writing and publishing scientific research articles in English and we believe that the revised manuscript with the latest proofreading meets the professional English language requirements.